# The Impact of Post-COVID-19 Syndrome on Self-Reported Physical Activity

**DOI:** 10.3390/ijerph18116017

**Published:** 2021-06-03

**Authors:** Jeannet M. Delbressine, Felipe V. C. Machado, Yvonne M. J. Goërtz, Maarten Van Herck, Roy Meys, Sarah Houben-Wilke, Chris Burtin, Frits M. E. Franssen, Yvonne Spies, Herman Vijlbrief, Alex J. van ’t Hul, Daisy J. A. Janssen, Martijn A. Spruit, Anouk W. Vaes

**Affiliations:** 1Department of Research and Development, Ciro, 6085 NM Horn, The Netherlands; FelipeMachado@ciro-horn.nl (F.V.C.M.); yvonnegoertz@ciro-horn.nl (Y.M.J.G.); maartenvanherck@ciro-horn.nl (M.V.H.); roymeys@ciro-horn.nl (R.M.); sarahwilke@ciro-horn.nl (S.H.-W.); fritsfranssen@ciro-horn.nl (F.M.E.F.); daisyjanssen@ciro-horn.nl (D.J.A.J.); martijnspruit@ciro-horn.nl (M.A.S.); anoukvaes@ciro-horn.nl (A.W.V.); 2NUTRIM School of Nutrition and Translational Research in Metabolism, 6229 HX Maastricht, The Netherlands; 3Department of Respiratory Medicine, Maastricht University Medical Centre (MUMC+), 6229 HX Maastricht, The Netherlands; 4REVAL—Rehabilitation Research Center, BIOMED—Biomedical Research Institute, Faculty of Rehabilitation Sciences, Hasselt University, 3590 Diepenbeek, Belgium; chris.burtin@uhasselt.be; 5Lung Foundation Netherlands, 3818 LE Amersfoort, The Netherlands; yvonnespies@longfonds.nl (Y.S.); hermanvijlbrief@longfonds.nl (H.V.); 6Department of Pulmonary Disease, Radboud University Medical Center, 6525 GA Nijmegen, The Netherlands; Alex.vantHul@radboudumc.nl; 7Department of Health Services Research, Care and Public Health Research Institute, Faculty of Health, Medicine and Life Sciences, Maastricht University, 6229 ER Maastricht, The Netherlands

**Keywords:** Corona, exercise, persistent symptoms, physical activity

## Abstract

*Background:* A subgroup of patients recovering from COVID-19 experience persistent symptoms, decreased quality of life, increased dependency on others for personal care and impaired performance of activities of daily living. However, the long-term effects of COVID-19 on physical activity (PA) in this subgroup of patients with persistent symptoms remain unclear. *Methods:* Demographics, self-reported average time spent walking per week, as well as participation in activities pre-COVID-19 and after three and six months of follow-up were assessed in members of online long-COVID-19 peer support groups. *Results:* Two hundred thirty-nine patients with a confirmed COVID-19 diagnosis were included (83% women, median (IQR) age: 50 (39–56) years). Patients reported a significantly decreased weekly walking time after three months of follow-up (three months: 60 (15–120) min. vs. pre-COVID-19: 120 (60–240) min./week; *p* < 0.05). Six months after the onset of symptoms walking time was still significantly lower compared to pre-COVID-19 but significantly increased compared to three months of follow-up (three months: 60 (15–120) min. vs. six months: 90 (30–150) min.; *p* < 0.05). *Conclusions:* Patients who experience persistent symptoms after COVID-19 may still demonstrate a significantly decreased walking time six months after the onset of symptoms. More research is needed to investigate long-term consequences and possible treatment options to guide patients during the recovery fromCOVID-19.

## 1. Introduction

At the time of writing, the number of people with a confirmed diagnosis of coronavirus disease 2019 (COVID-19) has risen to 167 million globally, with over 1.6 million confirmed cases in the Netherlands and over 1 million cases in Belgium [1]. As the pandemic continues, many patients have passed through the acute phase of the disease, but may still face difficulties in resuming their daily routine.

It has already been demonstrated that hospitalized patients with COVID-19 present with low physical functioning and impaired performance in activities of daily living (ADLs) immediately after discharge [2]. Several months after the infection, a subgroup of patients with COVID-19 still report ongoing symptoms such as fatigue, dyspnoea and muscle weakness, as well as impaired quality of life and increased dependency on others for personal care and the performance of ADLs [3,4,5,6,7,8,9,10,11]. This suggests the presence of a post-COVID-19 syndrome [4,12,13], which was recently referred to as post-acute sequelae of SARS-CoV-2 infection (PASC) by the National Institutes of Health (NIH) [14]. Moreover, a qualitative study indicated that the experience of prolonged symptoms hampers patients in resuming or maintaining physical activity [15]. Similar to other countries, the Dutch and Belgian governments decided to close sport clubs and requested residents’ home confinement as a measure to limit the rate of infections (Netherlands closed sport clubs on 15 March 2020 and requested home confinement on 23 March 2020; Belgium closed sport clubs and requested home confinement on 13 March 2020) [16,17]. Previous studies have already demonstrated that those governmental measures led to a decrease in physical activity (PA) among other unhealthy lifestyle behaviours in the general population [18,19]. However, the long-term effects in patients with COVID-19 on PA levels remain unclear.

Therefore, the aim of this study was to assess the impact of COVID-19 on the level of self-reported PA (time spent walking per week and leisure-time sports activities) in patients with post-COVID-19 syndrome (i.e., long COVID, long-haul COVID or PASC). We hypothesized a significant decrease in PA approximately three months after the infection, and at least a partial recovery of PA six months after the infection.

## 2. Materials and Methods

### 2.1. Study Design and Participants

As part of a large longitudinal study conducted in the Netherlands and Flanders (Belgium), an online questionnaire was made available between June 4 and June 11 of 2020 (T1) to all members of two long COVID Facebook groups [20,21] and to an online COVID-19 panel (www.coronalongplein.nl) (accessed on 4 June 2020). In total, 1556 participants who agreed to take part in a follow-up of this study were asked to complete a second survey between August 31 and September 8 of 2020 (T2). The medical ethics committee of the Maastricht University stated that the Medical Research Involving Human Subjects Act (WMO) did not apply to this study and that an official approval of this study by the committee was not required (METC2020-1978, METC2020-2554). The medical ethics committee of Hasselt University formally judged and approved the study (MEC2020/041). All adult respondents (aged 18 years or older) gave digital informed consent at the start of the questionnaires. Data about symptoms, care dependency, information needs, functional status, work productivity and quality of life have been published before [4,22,23,24,25,26].

### 2.2. Outcome Measures

The survey contained questions regarding demographics (sex (male/female/other), age (years), body mass index (BMI) (kg/m^2^), marital status (married or living with partner: yes/no), education level (low/medium/high)), self-reported pre-existing comorbidities, COVID-19 diagnosis (based on reverse transcription polymerase chain reaction (RT-PCR) test and/or computed tomography (CT) scan of the thorax; symptom-based medical diagnosis; no test/medical diagnosis), received care (no care needed/physiotherapy/rehabilitation), symptoms and admission to hospital.

In addition, participants were asked about the average time they spent walking in the previous seven days and which sports/activities they performed before COVID-19 (retrospectively) and at the time of completing the two questionnaires (approximately three and six months after symptom onset, respectively) (A summary of the questionnaire is added in the online supplement). The World Health Organization (WHO) recommends a minimum of 150 min per week of moderate-intensity aerobic physical activity [27]. Therefore, participants were divided into two groups: average walking time of less than 150 min per week or an average walking time of 150 min or more per week.

### 2.3. Statistical Analysis

Initial analyses were performed in participants with an RT-PCR or CT-confirmed diagnosis. Participants with a presumed COVID-19 diagnosis (*n* = 766; data presented in the online supplement) were excluded from the primary analyses.

Statistical analyses and visualizations were performed using SPSS v25.0 (IBM Corp., Armonk, NY, USA), SankeyMATIC (http://sankeymatic.com/build/) (accessed on 4 June 2020) and GraphPad Prism 8.3.5. (GraphPad Software, La Jolla, CA, USA). Data were presented as mean and standard deviation (SD), median and interquartile range (IQR) or frequency and proportion, as appropriate. Data were tested for normality with a Kolmogorov–Smirnov test. Within-group comparisons were performed using the Friedman test, McNemar’s test or standard Cochran’s Q test (with Bonferroni corrected post-hoc test). Between-group comparisons were performed using a Mann–Whitney U test or Fisher’s exact test. A priori, the level of significance was set at *p* < 0.05.

## 3. Results

### 3.1. Demographics and Characteristics

Of the initial 1556 participants that consented to be approached for a second questionnaire, 1005 participants completed the online questionnaires at approximately six months. See Appendix A for all details. Data from 239 participants with an RT-PCR or CT-confirmed diagnosis (82.8% women; median age: 50 (39–56) years) were used for the primary analyses, of which 62 (26%) were hospitalized (but not admitted to the intensive care unit (ICU)). The average time between the onset of symptoms and filling out the questionnaires was 10.4 ± 2.4 weeks (T1) and 22.6 ± 2.4 weeks (T2).

All characteristics of the 239 participants with a confirmed COVID-19 diagnosis are presented in Table 1. Results stratified for hospital admission are presented in the online supplement (Appendix A).

### 3.2. Self-Reported Walking Time

Results regarding self-reported time spent walking per week at the three time points (pre-COVID-19 (T0) and three (T1) and six months (T2) of follow-up) are presented in Figure 1. After three months of follow-up, walking time in the previous week was significantly reduced compared to pre-COVID-19 (three months: 60 (15–120) min. vs. pre-COVID-19: 120 (60–240) min./week; *p* < 0.05). Although there was a recovery in walking time between three months and six months of follow-up (from 60 (15–120) min. to 90 (30–150) min.; *p* < 0.05), walking time was still significantly lower compared to pre-COVID-19.

The proportion of participants reporting walking ≥150 min. per week at the different time points is presented in Figure 2. Approximately 41% of the participants reported walking ≥150 min. per week before COVID-19. This was significantly lower after three (17%) and six months (28%) of follow-up. Only 9% of participants reported walking ≥150 min. per week at all three time points. Approximately 20% of participants reported walking ≥150 min. per week pre-COVID-19 but did not achieve this at three and six months of follow-up, while 11% decreased their walking time per week (from ≥150 min. to <150 min. per week) at three months of follow-up but restored it to ≥150 min. at six months of follow-up. There were no significant differences in baseline characteristics between participants walking ≥150 min. per week and participants walking <150 min. per week, with the exception of the proportion of patients undergoing rehabilitation between three and six months of follow-up (<150 min.: 8%; ≥150 min.: 17%; *p* = 0.04). Additional analyses of walking time stratified for sex and number of symptoms are reported in the online supplements (Appendix A)

### 3.3. Activities

Participants reported a wide variety of activities, with walking (pre-COVID-19: 53.1%; three months: 41.1%; six months: 68.2%), outdoor cycling (pre-COVID-19: 35.1%; three months: 21.3%; six months: 42.3%) and (physio)fitness/exercise groups (pre-COVID-19: 30.1%; three months: 10.0%; six months: 38.5%) as the three most reported activities (Table 2).

At three months of follow-up, participants reported performing fewer activities compared to pre-COVID-19 and almost 44% of the participants were not able to be physically active or perform sports or activities due to COVID-19. From three months to six months of follow-up the proportion of participants unable to be physically active significantly decreased (from 44% to 12%; *p* < 0.05) and the proportion of participants reporting walking, cycling outdoors/indoors, participating in (physio)fitness/exercise groups and running significantly increased. The proportion of participants that reported walking and cycling indoors at six months was significantly higher compared to pre-COVID-19.

## 4. Discussion

This study aimed to assess the impact of COVID-19 on the level of self-reported PA (time spent walking per week and leisure-time sports activities) in patients with post-COVID-19 syndrome (i.e., long COVID, long-haul COVID or PASC). As hypothesized, a significant decrease in walking time approximately three months after the infection and a partial recovery of walking time six months after the infection was found.

These results indicate a possible recovery pattern similar to that which has been established in influenza A, acute respiratory distress syndrome survivors and severe acute respiratory syndrome, where patients experience impaired health-related quality of life, functional disability, psychological problems and impaired exercise capacity after up to two years of follow-up [28,29,30,31].

Since the Dutch healthcare system was completely overwhelmed by the COVID-19 pandemic, the primary focus was the treatment of patients with the most life-threatening symptoms. People without the need for ICU admission, were understandably not prioritized during these hectic times, and therefore not intensively monitored or guided in their recovery. Complications like myocarditis or thromboembolic problems might have been missed and therefore might have hindered the recovery of PA in this subgroup of patients [32,33].

Furthermore, since the effects of COVID-19 on the human body are still not completely clear, it is important to consider that patients might recover along different trajectories. Indeed, Gandotra et al. have investigated recovery in patients with respiratory failure due to several causes and identified different recovery trajectories and characteristics [34]. It is feasible that similar trajectories are present in patients recovering from COVID-19. Additionally, a study by Sallis et al. has shown that the level of PA pre-COVID-19 is associated with the severity of COVID-19. In fact, physical inactivity was the strongest risk factor for hospital admission, ICU admission and death, exceeding well-known risk factors like smoking, obesity, diabetes, hypertension, cardiovascular disease and cancer [35]. Since PA pre-COVID-19 seems to have a large impact on three important outcomes after COVID-19 (hospital admission, ICU admission and death), we recommend further investigation of the association between PA and outcomes after COVID-19 in a population of patients with persistent symptoms.

Although encouraging patients to return to performing daily activities and to start low/moderate-intensity exercise at home is currently recommended for patients recovering from COVID-19 [36,37], Humphreys et al. have described that patients experienced a lack of clear and consistent advice with regard to PA and refraining from PA after suffering a relapse in symptoms after PA or after seeing others relapse after PA [15]. In patients with persistent symptoms, it is recommended to perform a post-COVID-19 assessment and referring them to specialists or pulmonary rehabilitation based on the clinical findings [3]. A patient-tailored approach is needed in order to achieve an optimal recovery after an infection with COVID-19 [15].

While interpreting the abovementioned results, some strengths and limitations of this study need to be considered.

One of the strengths of this study is the fact that this is the first longitudinal study investigating PA in patients with post-COVID-19 syndrome. We were able to follow up with a significant number of participants during the course of six months. A team of scientists, methodologists and patients worked together during the preparation of this study in order to create a study that provided a complete overview of relevant parameters in this specific population.

However, since participants were recruited through platforms that targeted patients with persistent symptoms, there is the possibility of selection bias and the external validity of these results might be limited. It is possible that participants that recovered after three months did not feel the need to fill in the questionnaire at six months and therefore are underrepresented. However, the additional analyses demonstrate that non-responders had a significantly lower walking time three months after the onset of symptoms, compared to the responders (median (IQR) 40 (10–95) vs. 59 (15–105) min. respectively; *p* = 0.046) while walking time before COVID-19 was similar. This indicates that non-responders showed even less recovery of PA compared to responders.

Additionally, females are overrepresented in this sample, possibly due to the higher proportion of women that are part of online long COVID support groups [38]. This is consistent with the gender distribution in previous studies [8,38,39,40]. Additional analyses on walking times stratified for gender demonstrated that female participants had a significantly lower walking duration compared to males at T1 and T2, while no differences were found at T0. This could indicate that the recovery of walking time differs between males and females. Taking the previously mentioned limitations into consideration, there are limitations in extrapolating these results to the general population.

Due to the national regulations that were in place during the first wave of COVID-19 infections, the possibilities for sports and activity were limited. In both the Netherlands and Belgium, a lockdown was proclaimed, people were asked to stay at home as much as possible and sport facilities were closed. Therefore, the decrease in activities that were not possible during a large part of the three months after symptom onset (e.g., swimming, team sports) is probably caused by the regulations that were in place at that time. However, at the same time, many sports clubs initiated alternative options like online dance and exercise classes to provide the possibility to exercise at home or outdoors. The regulations that were in place at the time also do not explain the difference in walking time. However, the increase in the proportion of participants who reported walking as an activity might be related to the regulation.

Since the pandemic had a huge impact on healthcare resources, medical treatments were focused on the most severe cases of COVID-19 and patients with “milder” symptoms were often not tested. As a result, many patients who presented with milder symptoms were not admitted to the hospital or even officially diagnosed with COVID-19 and remained under the medical radar. The NICE guidelines have indicated that having a positive RT-PCR test or hospitalization is not a requirement for a COVID-19 diagnosis and healthcare should also focus on patients with suspected COVID-19 [13]. In line with these guidelines, we believe that the group of study participants with presumed COVID-19 provides valuable information and needs further attention. Results of participants with presumed COVID-19 are mostly comparable with the results of the participants with a confirmed diagnosis, and are presented in the online supplement (Appendix A).

Participants were asked to retrospectively assess their physical activity by filling in questionnaires. Therefore, the effect of recall bias cannot be excluded. However, since participants were asked questions that referred to the previous three months or the previous seven days, we expect that the influence of recall bias is limited. Besides recall bias in general, Dyrstad et al. have shown that a self-reported assessment of PA tends to overestimate vigorous activity and underestimate sedentary time [41]. This effect might have led to an overestimation of walking time in our results. Additionally, the questionnaire on sports and activities other than walking provided no information on frequency, intensity or duration, but only whether or not the activity was performed. This limitation should be kept in mind when interpreting these results. At the time of inclusion, due to the acute outbreak of COVID-19, it was impossible to provide participants with accelerometers and perform an objective pre-and post-COVID-19 PA assessment, but in future research objectively measured PA could be used to gain a more complete picture of the PA level in this population.

## 5. Conclusions

Participants with persistent symptoms while recovering from COVID-19 who were all members of online long COVID support groups still demonstrated a significantly decreased self-reported walking time six months after the onset of symptoms. In contrast, the proportion of participants that reported walking or cycling indoors increased over the course of six months after the onset of symptoms. More research is needed to investigate the long-term consequences and possible treatment options to guide patients during the recovery from COVID-19.

## Figures and Tables

**Figure 1 ijerph-18-06017-f001:**
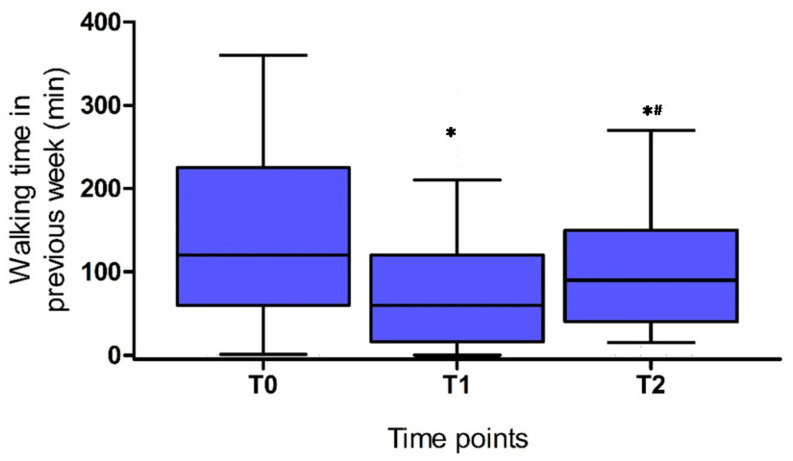
Walking time in previous week pre-COVID-19 (T0), at 3 months (T1) and 6 months (T2) of follow-up; * significant difference vs. pre-COVID-19, *p* < 0.05; ^#^: significant difference vs. 3 months, *p* < 0.05. Data presented as median, IQR and 10–90%CI.

**Figure 2 ijerph-18-06017-f002:**
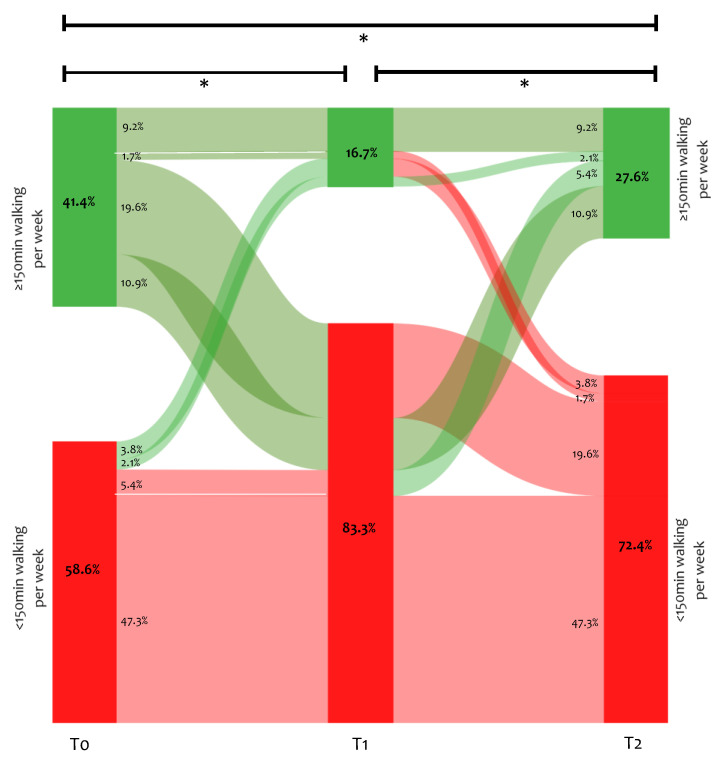
Proportion of participants divided based on achieving 150 min. walking per week before (T0) and at 3 months (T1) and 6 months (T2) of follow-up; * significant difference *p* < 0.05.

**Table 1 ijerph-18-06017-t001:** General characteristics of participants.

	All Participants (*n* = 239)
Hospitalized due to COVID-19, *n* (%)	62 (25.9)
Women, *n* (%)	198 (82.8)
Age, years, median (IQR)	50.0 (39.0–56.0)
Body mass index (BMI), kg/m^2^, median (IQR)	26.0 (23.4–30.5)
Marital status—married/living with partner, *n* (%)	173 (72.4)
Education level, *n* (%)	
Low	6 (2.5)
Medium	126 (52.7)
High	107 (44.8)
Self-reported pre-existing comorbidities, *n* (%)	
None	142 (59.4)
1 comorbidity	62 (25.9)
≥2 comorbidities	35 (14.6)
Received care	
No care needed, %	
Between infection and 3 months of follow-up	14.6
Between 3 and 6 months of follow-up	14.2
Physiotherapy, %	
Between infection and 3 months of follow-up	31.8
Between 3 and 6 months of follow-up	61.9 ^#^
Rehabilitation, %	
Between infection and 3 months of follow-up	4.2
Between 3 and 6 months of follow-up	11.7 ^#^

^#^*p* < 0.05 vs. 3 months of follow-up.

**Table 2 ijerph-18-06017-t002:** Activities performed by participants before (T0) and at three (T1) and six months (T2) of follow-up.

	Sport/Activity, *n* (%)	All Participants (*n* = 239)		
		T0	T1	T2
1	Walking	127(53.1)	99(41.1) *	163(68.2) *^#^
2	Cycling outdoors	84(35.1)	51(21.3) *	101(42.3) ^#^
3	(Physio)fitness/exercise groups	72(30.1)	24(10.0) *	92(38.5) ^#^
4	Swimming	24(10.0)	9(3.8) *	14(5.9) *
5	Running	24(10.0)	7(2.9) *	28(11.7) ^#^
6	Yoga/Pilates	12(5.0)	0(0.0) *	7(2.9)
7	Racket sports	10(4.2)	1(1.3) *	4(1.7) *
8	Team sports	8(3.3)	0(0.0) *	2(0.8)
9	Martial arts	5(2.1)	1(0.4)	0(0.0) *
	Cycling indoors	5(2.1)	10(4.2)	28(11.7) *^#^
	Work-related activities	5(2.1)	0(0.0) *	0(0.0) *
10	Dancing	4(1.7)	0(0.0) *	0(0.0) *
	No sports/activities, regardless of COVID-19	27(11.3)	9(3.8) *	5(2.1) *
	No sports/activities, due to COVID-19	N.A.	104(43.5)	28(11.7) ^#^

*: significant difference vs. pre-COVID-19, *p* < 0.05; ^#^: significant difference vs. three months, *p* < 0.05.

## Data Availability

The data presented in this study are available on reasonable request from the corresponding author, and only after approval by the medical ethical committee. The data are not publicly available due to ethical restrictions.

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
