# Peer review of "The Impact of Post-COVID-19 Syndrome on Self-Reported Physical Activity"

_ijerph, 2021, doi:10.3390/ijerph18116017_

Round 1
Reviewer 1 Report
After reading this manuscript with great interest, I believe that this article could be accepted despite limitations regarding the measurement of physical activity. Indeed, the measurement of physical activity is subjective and doesn't allow to know the characteristics of the physical activity practiced.
Author Response
Dear Editor in Chief,
We hereby submit a detailed point-by-point reply and the revised version of our manuscript ijerph-1226919 titled "The impact of the post-COVID-19 syndrome on self-reported physical activity".
We kindly ask you to take our manuscript into consideration for publication in International Journal of Environmental Research and Public Health.
On behalf of all authors,
Sincerely,
Ms. Jeannet M. Delbressine
C1. After reading this manuscript with great interest, I believe that this article could be accepted despite limitations regarding the measurement of physical activity. Indeed, the measurement of physical activity is subjective and doesn't allow to know the characteristics of the physical activity practiced.
R1. Thank you for reviewing our manuscript. We agree that an objective measurement of physical activity would significantly improve the research design. However, at the time the study was conducted this was not possible. We hope to use objectively measurements of physical activity in future research designs. This is also mentioned in the manuscript (Line 344-349)

Reviewer 2 Report
Given the current nature of the COVID-19 pandemic, this is a timely study that examines how persistent post-infection symptoms affect self-reported walking time. Although the study is limited based on its nature using self-reported and recalled activity, it fills an important gap in the literature.
Major comments:
- The manuscript needs to undergo careful English editing. For example, there is a comma error in the first sentence. In line 29, the comma should precede, rather than follow, "and". In addition, the sentence on line 35 starts with a numeral. Numbers should be spelled out when starting a sentence. While pointing out every error is beyond the scope of this review, the entire manuscript requires polishing before it is publication-ready, as there are other punctuation and syntax errors in the paper.
- Please break the introduction into three paragraphs. The first paragraph should introduce the topic, the second paragraph should provide the background information, and the third paragraph should provide the aims of the study. This will greatly improve the readability of the introduction.
- While the methods are well-described, I feel that this paper could benefit from additional analyses.
- Table 1 presents the participant characteristics. Were there any differences in any of these characteristics in the subgroups (< or >=150 minutes per week)? This is important as it may affect the overall results. In addition, any future studies that seek to apply a multivariate or multivariable analysis could draw upon this information for potential confounders.
- How does stratifying the results by sex affect the analysis? As the overwhelming majority of participants were female, does this perhaps negate potential differences in male participants?
- Regarding the study limitations, is it also possible that non-responders for the last survey are those who are least motivated both to respond and to engage in physical activity? It seems that these two behaviors could be linked, and this could explain why non-responders tended to have less physical activity at T1. Did these same participants also have low physical activity at T0?
- Although the discussion is already an appropriate length, I feel it could benefit by adding any similar information that has been derived from other illnesses such as the flu. Are there any data available following recovery from other illnesses that can be used for comparison?
Minor comments:
- Title for Table 2 is incomplete. Shouldn't this end with "up"?
Author Response
Dear Editor in Chief,
We hereby submit a detailed point-by-point reply and the revised version of our manuscript ijerph-1226919 titled "The impact of the post-COVID-19 syndrome on self-reported physical activity".
We kindly ask you to take our manuscript into consideration for publication in International Journal of Environmental Research and Public Health.
On behalf of all authors,
Sincerely,
Ms. Jeannet M. Delbressine
C1. The manuscript needs to undergo careful English editing. For example, there is a comma error in the first sentence. In line 29, the comma should precede, rather than follow, "and". In addition, the sentence on line 35 starts with a numeral. Numbers should be spelled out when starting a sentence. While pointing out every error is beyond the scope of this review, the entire manuscript requires polishing before it is publication-ready, as there are other punctuation and syntax errors in the paper.
R1. We have made adjustments throughout the manuscript in order to remove punctuation and syntax errors and have asked a native English-speaking colleague to check the text and correct mistakes.
C2. Please break the introduction into three paragraphs. The first paragraph should introduce the topic, the second paragraph should provide the background information, and the third paragraph should provide the aims of the study. This will greatly improve the readability of the introduction.
R2. Thank you for this suggestion. We have made the suggested changes in the introduction.
C3. While the methods are well-described, I feel that this paper could benefit from additional analyses.
Table 1 presents the participant characteristics. Were there any differences in any of these characteristics in the subgroups (< or >=150 minutes per week)? This is important as it may affect the overall results. In addition, any future studies that seek to apply a multivariate or multivariable analysis could draw upon this information for potential confounders.
R3. Based on the suggestions of the reviewer, we have performed an additional analyses of the general characteristics stratified for below 150 min. of walking per week and equal or above 150 min. walking per week and found no significant differences with the exception of the proportion of patients who underwent rehabilitation between 3 and 6 months of follow up(< 150 min.: 8% vs. ≥150 min. 17%; p=.04). We have added this to the manuscript in line 183-189.
C4. How does stratifying the results by sex affect the analysis? As the overwhelming majority of participants were female, does this perhaps negate potential differences in male participants?
R4. Based on the suggestions of the reviewer, we have performed an additional analysis for walking duration stratified for sex. We have found significantly lower walking times for females at both T1 and T2 (T1 male 90[37.5-180] min. vs female 50[10-105] min., p=.001 and T2 male 120[60-240] min. vs female 80[30-142.5], p=.003) and have added these results to the online supplement (Supplementary table 4). We have also mentioned this in the discussion (line 298-304):
“Additional analyses on walking times stratified for gender demonstrated that female participants had a significantly lower walking duration compared to males at T1 and T2, while no differences were found at T0. This could indicate that the recovery of walking time differs between males and females.”
C5. Regarding the study limitations, is it also possible that non-responders for the last survey are those who are least motivated both to respond and to engage in physical activity? It seems that these two behaviors could be linked, and this could explain why non-responders tended to have less physical activity at T1. Did these same participants also have low physical activity at T0?
R5. Indeed, it is reasonable to assume that the non-responders are those least motivated to respond and may also engage to a lesser extent in physical activity. However, we have already performed this analysis and compared the responders and non-responders’ PA at T0 and found no significant differences as mentioned in line 289-293: “However, the additional analyses demonstrate that non-responders had a significant lower walking time three months after the onset of symptoms, compared to the responders (median [IQR] 40[10-95] vs 59[15-105] min., respectively; p=0.046) while walking time before COVID-19 was similar.”
C6. Although the discussion is already an appropriate length, I feel it could benefit by adding any similar information that has been derived from other illnesses such as the flu. Are there any data available following recovery from other illnesses that can be used for comparison?
R6. Thank you for this suggestion. We have added results from studies in patients with SARS, ARDS and influenza A with a two year follow up in line 234-239:
“These results indicate a possible similar recovery pattern that has been established in Influenza A, acute respiratory distress syndrome survivors and severe acute respiratory syndrome. Patients experienced impaired health-related quality of life, functional disability, psychological problems and impaired exercise capacity up to two years of follow up28-31 “
C7. Title for Table 2 is incomplete. Shouldn't this end with "up"?
R7. The word “up” is in the title (line 201), however during the editing process this word has been placed on the following line. If the manuscript will be published we will make sure this is adjusted.

Reviewer 3 Report
It would be very important to use objective measures with accelerometers to assess physical activity in order to confirm the results of this study in the post-covid-19 patient population at different periods after infection.
Author Response
Dear Editor in Chief,
We hereby submit a detailed point-by-point reply and the revised version of our manuscript ijerph-1226919 titled "The impact of the post-COVID-19 syndrome on self-reported physical activity".
We kindly ask you to take our manuscript into consideration for publication in International Journal of Environmental Research and Public Health.
On behalf of all authors,
Sincerely,
Ms. Jeannet M. Delbressine
C1. It would be very important to use objective measures with accelerometers to assess physical activity in order to confirm the results of this study in the post-covid-19 patient population at different periods after infection.
R1. Thank you for reviewing our manuscript. We agree that an objective measurement of physical activity would significantly improve the research design. However, at the time the study was conducted this was not possible. We hope to use objectively measurements of physical activity in future research designs. This is also mentioned in the manuscript (Line 344-349)

Reviewer 4 Report
- Can authors do subgroup analysis according to disease severity (asymptomatic, mild and moderate)?
- Selection bias is possible. People who are willing to participate in the study are healthier. Moreover, the majority of participants are female. Extrapolation will be limited.
- Would the level of government's epidemic prevention measures affect people's activities?
- No objective pre-and post-COVID-19 health assessment for each participant is a major limitation.
Author Response
Dear Editor in Chief,
We hereby submit a detailed point-by-point reply and the revised version of our manuscript ijerph-1226919 titled "The impact of the post-COVID-19 syndrome on self-reported physical activity".
We kindly ask you to take our manuscript into consideration for publication in International Journal of Environmental Research and Public Health.
On behalf of all authors,
Sincerely,
Ms. Jeannet M. Delbressine
C1. Can authors do subgroup analysis according to disease severity (asymptomatic, mild and moderate)?
R1. Unfortunately we do not have data on disease severity. We do have data on the number of symptoms, but not on the severity of those symptoms. Based on the suggestion of the reviewer we have performed an additional analysis based on the number of symptoms during the acute phase of the disease and have added this to the online supplement (Supplementary table 5). These analyses demonstrated that there were significant differences in walking time between the group with 6-10 symptoms and >10 symptoms at T1 and T2 (T1: 6-10 symptoms: 120[60-195] min. vs >10 symptoms: 40[10-100] min., p=.000 and T2 6-10 symptoms: 120[60-255] min. vs >10 symptoms: 77.5[32.75-140] min., 0=.010)
C2. Selection bias is possible. People who are willing to participate in the study are healthier. Moreover, the majority of participants are female. Extrapolation will be limited.
R2. We agree with the reviewer, and added the selection bias in the section were we describe the limitations in external validity (“However, since participants were recruited through platforms that targeted patients with persistent symptoms, there is the possibility of selection bias and the external validity of these results is limited.” Line 284-287). We have already described that females are overrepresented (line 295-297). We have added the following: “Taking the previous mentioned limitations into consideration, there might be limitations in extrapolating these results to the general post-COVID population.”(line 302-304).
C3. Would the level of government's epidemic prevention measures affect people's activities?
R3. Indeed, national regulations that were in place during the first wave of COVID-19 infections could have affect the level of physical activity. This is already mentioned in the discussion (line 305-318).
C4. No objective pre-and post-COVID-19 health assessment for each participant is a major limitation.
R4. Thank you for reviewing our manuscript. We agree that an objective measurement of physical activity would significantly improve the research design. However, at the time the study was conducted this was not possible. We hope to use objectively measurements of physical activity in future research designs. This is also mentioned in the manuscript (Line 344-349)

Reviewer 5 Report
Comments to the Author:
I thank to the editors for the opportunity to review this study, beside I would also like to congratulate the authors for the made effort in their study. The present manuscript by Delbressine et al., analyzed “The impact of the post-COVID-19 syndrome on self-reported physical activity”, the authors attempted to assess the impact of COVID-19 on the level of self-reported physical activity (time spent walking per week and leisure-time sports activities) in patients with post-COVID-19 syndrome (i.e. long COVID, long-haul COVID patients or PASC). The manuscript lacks novelty, since it is understandable that subjects suffering from COVID-19 will not recover their health or physical condition, at least after the disease has passed. It was therefore to be expected to find a drastic reduction in routine walking. In additions, some issues need to be addressed to improve the presentation of the study.
- Authors must complete the introduction section, given than I am missing plenty information about improvements in physical activity, the initial date of quarantine in Netherlands, more information about the other countries that have closed their borders, how quarantine status (extreme sedentary lifestyle) could affect the population, etc.
- Could the authors justify why there were so many differences between the number of male and female participants?
- What type of questionnaire was used? Was it validated? Was it validated in the language of the surveyed children? Authors should provide the name of the questionnaires through different citations.
- The authors have decided to investigate an important topic also mentioned as a hot topic nowadays, however in the supplementary material I didn’t find the questionnaire or even a summary in the questionnaire.
- The first paragraph of the discussion should be much better structured. First it should be the main aim of the study and then the most relevant results. It is not necessary to add more information because it confuses the reader.
- The authors should be clearer about the strengths of their study.
Author Response
Dear Editor in Chief,
We hereby submit a detailed point-by-point reply and the revised version of our manuscript ijerph-1226919 titled "The impact of the post-COVID-19 syndrome on self-reported physical activity".
We kindly ask you to take our manuscript into consideration for publication in International Journal of Environmental Research and Public Health.
On behalf of all authors,
Sincerely,
Ms. Jeannet M. Delbressine
C1. Authors must complete the introduction section, given than I am missing plenty information about improvements in physical activity, the initial date of quarantine in Netherlands, more information about the other countries that have closed their borders, how quarantine status (extreme sedentary lifestyle) could affect the population, etc.
R1. Thank you for this suggestion. We have added information on the effects of governmental COVID-19 measures in the general population and the dates of the start of home confinement and closing sport clubs in the Netherlands and Belgium in the introduction:
“Similar to other countries, the Dutch and Belgium government decided to close sport clubs and requested home-confinement as a measure to limit the rate of infections (Netherlands: 15 March 2020: closing sport clubs, 23 March 2020: Request of home confinement; Belgium: 13 March 2020: Closing sport clubs and request of home confinement 16,17. Previous studies have already demonstrated that those governmental measures have led to a decrease in physical activity (PA) among other unhealthy lifestyle behaviours in the general population 18,19 (line 68-76)
C2. Could the authors justify why there were so many differences between the number of male and female participants?
R2. As described in the paragraph starting in line 295, it is known that females are overrepresented in online long-COVID peer support groups.
C3. What type of questionnaire was used? Was it validated? Was it validated in the language of the surveyed children? Authors should provide the name of the questionnaires through different citations.
R3. No children were surveyed in this study as the minimal age for participation was 18 years. The questions related to the walking duration and activities were not validated specifically, but were based on suggestions of scientists, methodologists, healthcare professionals and COVID-19 patients from the Facebook groups of The Netherlands and Flanders. The questions have been added to the online supplement:
Sports/activities:
- T0: Before the Corona infection I have performed the following sport or activities on a weekly basis.
- T1 and T2: During the past week I have performed the following sports or activities.
- Participants were asked to select all answer options that applied and we added an open text field in case the activity or sport was not mentioned in the answer options.
Walking:
- T0:
- Before the Corona infection I walked XX times per week.
- On average a walk/hike took XX minutes
- T1 and T2
- During the past week I walked XX times
- On average a walk/hike took XX minutes
C4. The authors have decided to investigate an important topic also mentioned as a hot topic nowadays, however in the supplementary material I didn’t find the questionnaire or even a summary in the questionnaire.
R4. Scientists, methodologists, healthcare professionals and COVID-19 patients from the Facebook groups of The Netherlands and Flanders were closely involved in preparation of this study and development of the questionnaire. Based on the suggestion of patient representatives we have decided to limit the amount of questions in order to keep it feasible for participants to fill in the questionnaires. The questions, which are also mentioned in the response to the previous comment, have been added to the online supplement.
C5. The first paragraph of the discussion should be much better structured. First it should be the main aim of the study and then the most relevant results. It is not necessary to add more information because it confuses the reader.
R5. We adjusted the first paragraph of the discussion in which we now repeat the aim and mention the most relevant results. We have removed additional information from this paragraph.
The first paragraph of the discussion now states:
“This study aimed to assess the impact of COVID-19 on the level of self-reported PA (time spent walking per week and leisure-time sports activities) in patients with post-COVID-19 syndrome (i.e. long COVID, long-haul COVID patients or PASC). As hypothesized a significant decrease in walking time approximately three months after the infection and a partial recovery of walking time six months after the infection was found.”
C6. The authors should be clearer about the strengths of their study.
R6. We have added strengths of the study in the discussion in line 277-283:
“One of the strengths of this study is the fact that this is the first longitudinal study investigating PA in patients with Post-COVID-19 syndrome. We were able to follow up a significant number of participants during the course of six months. A team of scientists, methodologists and patients worked together during the preparation of this study in order to create a study that provided a complete overview of relevant parameters in this specific population.”

Round 2
Reviewer 5 Report
For the author:
I appreciate authors’ effort. The authors have obviously spent considerable time revising the manuscript and their hard work is clearly paying off. This manuscript is drastically improved from the original submission. The message is very clear, the language is much more clean, and the issues in the first version were corrected. Besides, the authors have answered all my comments successfully. For this reason, I encourage to editor to consider this manuscript for publication for the interesting value of the study realized, that now it is a much more robust study.